# The Identification of Potential Immunogenic Antigens in Particular Active Developmental Stages of the Rice Weevil (*Sitophilus oryzae*)

**DOI:** 10.3390/ijerph20053917

**Published:** 2023-02-22

**Authors:** Joanna Witecka, Natalia Malejky-Kłusek, Krzysztof Solarz, Olga Pawełczyk, Małgorzata Kłyś, Aleksandra Izdebska, Weronika Maślanko, Marek Asman

**Affiliations:** 1Department of Parasitology, Faculty of Pharmaceutical Sciences in Sosnowiec, Medical University of Silesia, 40-055 Katowice, Poland; 2Chair of Ecology and Environmental Protection, Institute of Biology, Pedagogical University of Cracow, Podchorążych 2 St., 30-084 Cracow, Poland; 3Department of Animal Ethology and Wildlife Management, Faculty of Animal Sciences and Bioeconomy, University of Life Sciences in Lublin, Akademicka 13 St., 20-950 Lublin, Poland; 4Department of Medical and Molecular Biology, Faculty of Medical Sciences in Zabrze, Medical University of Silesia, 40-055 Katowice, Poland

**Keywords:** *Sitophilus oryzae*, rice weevil, storage insects, allergy, sensitization

## Abstract

Background: The rice weevil (*Sitophilus oryzae*) originates from subtropical and tropical areas of Asia and Africa, but it also appears on other continents, mostly as a result of trade in rice. It may occur in grain fields as well as in storage facilities, and cause allergenic reactions. The aim of this study was to identify the potential antigens in all developmental stages of *S. oryzae,* which may cause an allergic response in humans. Methods: Sera of 30 patients were tested for the presence of IgE antibodies to antigens from three life stages of the rice weevil. To identify protein fractions containing potential allergens, proteins collected from larvae, pupae, and adults separated by sex of *S. oryzae* were fractionated by SDS-PAGE. Then, they were probed with anti-human, anti-IgE monoclonal antibodies, fractionated by SDS-PAGE and detected by Western blotting. Results: In total, 26 protein fractions of males and 22 fractions of other life stages of *S. oryzae* (larvae, pupae, and females) positively reacted with the examined sera. Conclusions: The conducted study showed that *S. oryzae* may be a source of many antigens which may cause the potential allergic reactions in humans.

## 1. Introduction

Pests are organisms that pose hazards to human health, well-being, and property [1]. They serve as vectors for pathogens and compete with humans for food and shelter [2,3]. This group of organisms also includes insects and mites that occur in stored products of plant and animal origin, which are referred to as storage pests. Such pests cause substantial quantitative and qualitative losses to the products on which they feed, and in which they live. Moreover, they contaminate foodstuffs with their bodies, excrement, and exuviae and cause an increase of humidity, because of heating the products in which they live. The storage mites are inhaled by the grain storage workers, irritate their mucous membranes, and lead to asthma [1]. Mites also attack the skin of food storage employees, causing its redness, eczema, and itching [4,5,6,7]. Similar to stored mites, the stored insects also can cause human allergies [8]. Consumption of flour products containing milled insect cuticles may result in inflammation and intestinal disorders in people and animals [5,9,10,11,12,13,14,15].

The rice weevil, *Sitophilus oryzae* (L.) (Coleoptera: Curculionidae) is a small (2.5–4 mm body length), dark reddish-brown beetle. It has a long snout characteristic of weevils [16]. *S. oryzae* originates from subtropical and tropical areas of Asia and Africa and has spread throughout the globe mostly as a result of trade in rice [17]. This beetle species is a major grain pest in Australia, Mexico, Africa, India, and south-eastern Asia [18,19,20]. It may occur in grain fields as well as in storage facilities. In temperate climate zones, it can survive only in heated food storage premises such as flour mills and warehouses [21]. The rice weevil is polyphagous, and it can feed on grains of rice, corn, buckwheat, sorghum, cereals, dried fruit, flour products, nuts, and tobacco [16,22,23,24,25,26,27]. Furthermore, this species may be allergenic for humans and it may play an important role as the cause of allergy in bakery employees, which is considered an occupational disease [1]. The main symptoms of allergy for *S. oryzae* belong to atopic rhinitis and bronchial asthma [6,8,28,29].

Therefore, the aim of this study was to identify the potential antigens in all active developmental stages of *S. oryzae* which may cause an allergic response in humans.

## 2. Materials and Methods

The examined specimens of *S. oryzae* were bred in the laboratory of the Chair of Ecology and Environmental Protection (Institute of Biology, Pedagogical University of Cracow), at conditions of 29 ± 1 °C with 60 ± 5% relative humidity (RH). The 15-day-old, adult beetles of *S. oryzae* used in the tests were obtained from breeding colonies kept in incubators (type C-100G) in dark. Wheat grain was the food and place for egg laying for beetles [30]. In turn, using two dissecting needles, *S. oryzae* larvae were isolated directly from wheat grains under a binocular magnifier in a Petri dish. Then, their body length was measured and on this basis, the third larval stage was determined and selected. Whole body extracts were prepared from larvae, pupae, males, and females of the examined insect species. We used third instar of the larvae. To prepare homogenates, the five individuals of each life stage were used [30]. The insect bodies were suspended in 1× Sample Buffer (SB), homogenized, and centrifuged at 12,000 rpm for 2 min in the laboratory centrifuge MPW 120 (MPW, Warsaw, Poland). The supernatant was carried to the new sterile 1.5 mL tube. For the presence of antigens, a total of 30 randomly selected human sera from the suburban population of Upper Silesia (South Poland) were examined. The homogenates were incubated at 100 °C for 5 min. Next, samples were centrifuged at 12,000 rpm for 2 min in the laboratory centrifuge MPW 120 (MPW, Poland). The SDS-PAGE was performed according to Laemmli [31] with some modifications, using a mini-Protean II system (BioRad, Hercules, CA, USA). The samples were separated electrophoretically in 12% polyarcrylamide gels at 100 V for 90 min. Then, an electrotransfer at 150 mA for one hour by a modified method of Towbin et al. [32] using a Mini-Transblot Cell (BioRad, USA) was performed. After this step, nitrocellulose membranes were blocked with skim milk (Sigma-Aldrich, Taufkirchen, Germany) and incubated at 4 °C with 150 µL of human sera diluted 1:100 in 1× Tris Buffered Saline (TBST) for about 16 h. Next, membranes were washed 3 times for 15 min in 1× suspended TBST Buffer. Then, samples were incubated for 2 h at ambient temperature diluted 1:1000 in 1× TBST Buffer anti-human IgE (Sigma-Aldrich, Germany). After this step, samples were washed 3 times for 15 min in 1 time suspended TBST Buffer and then in AP Buffer for 5 min. Then, a 7 mL BCIP/NBT Liquid Substrate System (Sigma-Aldrich, Germany) was added to the nitrocelluloses. The incubation was performed for 20 min in the dark at room temperature. Next, samples were washed in Stop Buffer for 30 s. Then, nitrocellulose membranes were dried in the air and photographed using the device Omega 10 (UltraLum, Berlin, CT, USA). For the analysis, the Total Lab Software was used (TotalLab, Gosforth, UK). The statistical analysis was performed using CSS-Statistica for Windows version 12. Statistical significance was declared at a *p* value of less than 0.05. Results were analyzed using the Yates corrected χ^2^ test.

## 3. Results

In total, 26 protein fractions with positive reactions for the examined sera were showed in males of *S. oryzae*. In larvae, pupae, and females a total of 22 fractions were reported (Figure 1, Figure 2, Figure 3 and Figure 4). In the case of females, the most sera (83.9%) reacted with the protein fractions of the size 33–36 kDa. In turn, fractions of the size 18, 67, 78, 173, 202, and 219 kDa gave significantly less frequent positive reactions (3.2%) with the studied sera (Yates corrected χ^2^ = 130.2; *p* ≤ 0.00001) (Figure 1).

Moreover, the protein fractions of 33–37 kDa in males of *S. oryzae* gave positive reactions with 80.6% of the examined sera, whereas the protein fractions present in males of 3, 7, 44, 70, 79, 113, 121, 173, 213, and 267 kDa reacted only with 3.2% of the sera (Figure 2). This difference was also statistically significant (Yates corrected χ^2^ = 121.7; *p* ≤ 0.00001).

The protein fractions present in *S. oryzae* pupae of 18–22, 23–26, and 45–53 kDa gave the most positive reactions with the examined sera, at 87.1%, 83.9%, and 83.9% of the sera, respectively (Figure 3). This difference is statistically not significant (Yates corrected χ^2^ = 0.16; *p* = 0.6879). Moreover, two other protein fractions of 68–76 kDa and 33–36 kDa were showed in the examined pupae. These fractions also showed high frequencies of positive reactions with the examined sera, namely 67.7% and 58.1%, respectively (Figure 3). This difference is also statistically insignificant (Yates corrected χ^2^ = 1.74; *p* = 0.1875). The remaining protein fractions on pupae of 114, 152, 176, 214, 249, 260, and 273 kDa significantly less frequently gave positive reactions (only 3.2% of sera, Figure 3) (Yates corrected χ^2^; *p* ≤ 0.00001 in all cases).

In the case of larvae, the largest quantity of sera (87.1%) positively reacted with protein fractions of the size 21–23 kDa (Figure 4). Moreover, the protein fractions of the size 20 kDa and 15–17 kDa present in this developmental stage gave positive reactions with over 50% of the examined sera, at 58.1% and 51.6%, respectively (Figure 4), and both differences were statistically significant (Yates corrected χ^2^ = 19.66 and 27.27, respectively; *p* ≤ 0.00001 in both cases). However, the difference between fractions 20 kDa and 15–17 kDa was statistically insignificant (Yates corrected χ^2^ = 0.51; *p* = 0.4773). In turn, the significantly smallest quantity of sera (3.2%) gave positive reactions with the protein fractions of the size 88, 126, 134, 158, 286 kDa (Figure 4) (Yates corrected χ^2^; *p* ≤ 0.00001 in all cases).

## 4. Discussion

It is commonly known that pest arthropods contaminate an environment by allergens and pathogens [13,33,34,35,36]. The allergic factors derived from these arthropods may cause occupational diseases like eczema or itching in warehouse workers [8]. Moreover, the mite and insect allergens can induce many burdensome disease entities in humans, such as conjunctivitis, allergic rhinitis, atopic dermatitis, and asthma [35].

The study conducted in the Czech Republic by Stejskal and Hubert [13] showed the occurrence of many pest arthropod species in grain stores. Apart from mites and psocids, the researchers found approximately 35,000 individuals of beetles in the samples collected from the grain stores. They reported the presence of 32 species of beetles, including *S. oryzae*, *Sitophilus granarius* (L.) (Coleoptera: Curculionidae), *Oryzaephilus surinamensis* (L.) (Coleoptera: Silvanidae), among others [13]. In turn, Jakubas-Zawalska et al. [17,37] showed many potential antigens in *S. granarius* and *O. surinamensis*, which may cause an allergic response in humans. The number of identified antigens in both species was various, and depended on the insect’s developmental stage. In molecular analysis of sera with *S. granarius* body extracts, the largest number of antigens in pupae, and the smallest in larvae was shown [37], whereas in the case of *O. surinamensis* the largest number of proteins with the allergenic potential was presented in females and pupae, and the smallest in larvae and males [17]. In the presented work, the largest number of protein fractions with an allergenic potential in males of *S. oryzae* was detected, while the smallest number of antigens in females, larvae, and pupae of this storage insect was shown. Moreover, this study shows that the protein fractions of each studied developmental stage of *S. oryzae* reacted with the IgE antibodies of the analyzed sera, with a various frequencies. A similar relationship was presented in the human immunological response to the antigens of *S. granarius* and *O. surinamensis* [17,37].

Kleine-Tebbe et al. [38] analyzed IgE-mediated inhalant allergy in inhabitants of buildings infested with *S. oryzae.* Their study showed that the protein fractions of the size 35–38, 54, 67, 70, and >94 kDa were binding to human IgE and may be potential allergens. In the presented study, the protein fractions at the same or similar locations were detected. Moreover, in comparison with the results obtained by Kleine-Tebbe et al. [38], in the present work a lot of other antigens in *S. oryzae* at locations below 35–38 and above 94 kDa were stated.

## 5. Conclusions

The conducted study showed that *S. oryzae* may be the source of many antigens which may cause potential allergic reactions in humans. Moreover, the obtained results confirmed that, similar to the case of *S. granarius* and *O. surinamensis*, each of the analyzed developmental stages of *S. oryzae* is characterized by the domination of the potential allergenic protein fractions with various molecular sizes. Due to the fact that storage beetles are widespread and given their potential allergenic significance, more attention should be paid to the human risk of exposure to contact with these insects and their antigens [13]. This may apply to both direct and indirect contact with infested stored products, organic dust, and food. In order to protect employees who handle stored products or work in warehouses, it is recommended to strengthen preventive measures regarding hygiene procedures in warehouses where cereals and their milling products are stored. In addition, preventive measures are recommended for products brought to warehouses and wholesalers, such as quarantining these products before placing them in warehouses and taking random samples of these products for analysis by specialists for the presence of insects and mites of warehouse pests. In addition, employees who handle stored products or work in warehouses should wear protective clothing, gloves and work masks to avoid allergic reactions to insects and storage mites.

## Figures and Tables

**Figure 1 ijerph-20-03917-f001:**
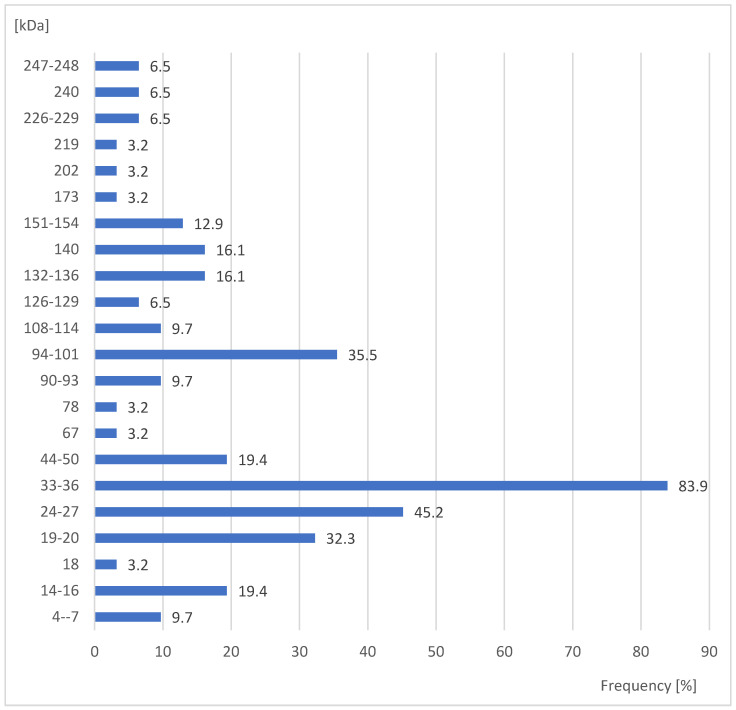
Percent of studied sera giving positive reactions with antigens in the extracts of females *Sitophilus oryzae*.

**Figure 2 ijerph-20-03917-f002:**
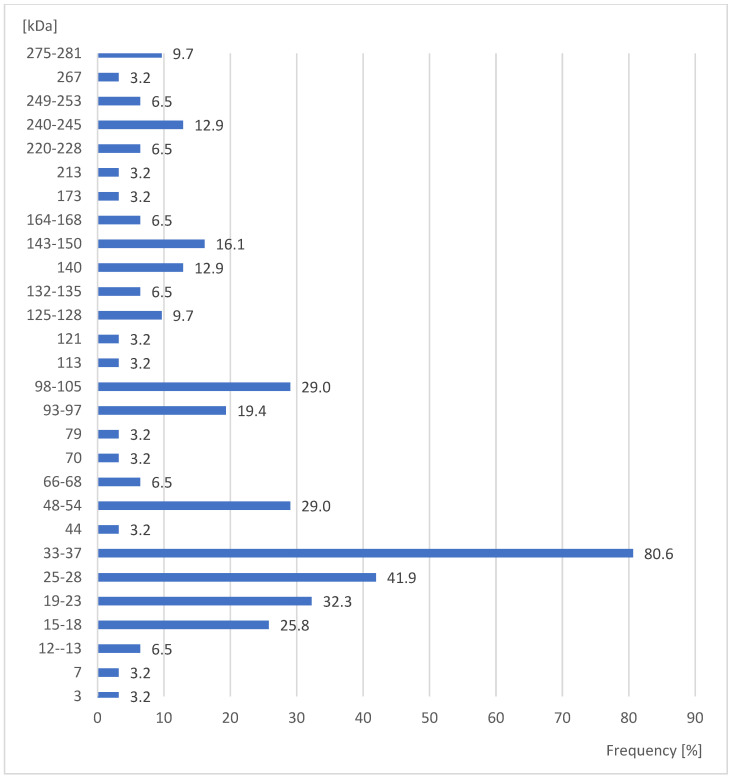
Percent of studied sera giving positive reactions with antigens in the extracts of males *Sitophilus oryzae*.

**Figure 3 ijerph-20-03917-f003:**
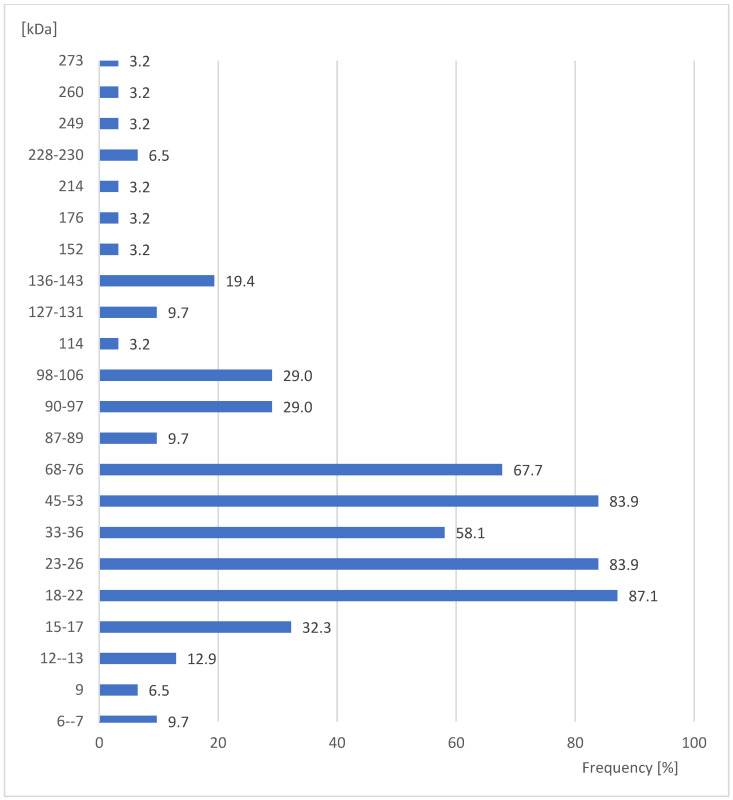
Percent of studied sera giving positive reactions with antigens in the extracts of pupae *Sitophilus oryzae*.

**Figure 4 ijerph-20-03917-f004:**
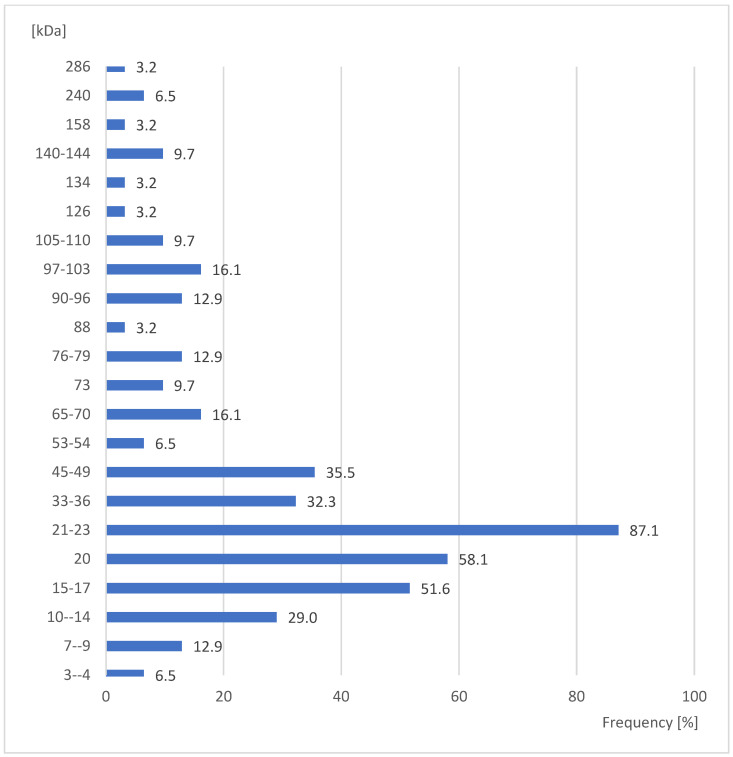
Percent of studied sera giving positive reactions with antigens in the extracts of *Sitophilus oryzae* larvae.

## Data Availability

Not applicable.

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
