# Peer review of "The Identification of Potential Immunogenic Antigens in Particular Active Developmental Stages of the Rice Weevil (Sitophilus oryzae)"

_ijerph, 2023, doi:10.3390/ijerph20053917_

Round 1

Reviewer 1 Report

Dear authors,

The manuscript is very interesting but you need to fix some points, as well as the format of the references. Please find my comments in the attached pdf file.

Author Response

Reviewer 1

Thank you very much to the reviewer for constructive comments on our manuscript.

All the remarks from the Reviewer were taken into account.

We added and removed commas in abstract.

We added references in several places in the text.

We wrote “The rice weevil (Sitophilus oryzae) as: "The rice weevil, Sitophilus oryzae (L.) (Coleoptera: Curculionidae)”.

In Materials and Methods we wrote at what temperature, reletive humidity, commodity and light we cultured S.oryzae. We added information what age were the adults and what was the instar of the larvae that was used.

These 30 samples of human serum are the minimum to perform statistical analysis with the use of the Yates corrected  χ2  test and obtain statistical significance of the test. Due to the nature of this work, which aim was to identify the potential antigens in all active developmental stages of S. oryzae wich may be potentially allergenic to humans demographic data were insignificant and therefore they were omitted.

The Yates corrected  χ2  test  is one of the most commonly used and widely accepted statistical test in this type of studies. Therefore the name of this test and correction is in our opinion enough as reference. The original paper of this test’s correction is from the year 1934 (https://doi.org/10.2307/2983604).

We provided authorities, order and family for each insects species.

We added "Conclusions" as a separate part in the manuscript.

We fixed reference format proper for the journal.

Reviewer 2 Report

The manuscript identified protein fractions in S. oryzae that caused an allergic response in humans. The result is not surprising, but it contributes to a body of knowledge that highlights the need to protect food products in storage from S. oryzae and other commonly occurring insect pests.

Some grammatical corrections are suggested at the following lines:

Line 146 This sentence is confusing.

Line 165 delete “were”

Line 170 “buildings”

Line 174 “present”

Author Response

Reviewer 2

Thank you very much to the reviewer for constructive comments on our manuscript.

All the remarks from the Reviewer were taken into account.

Line 146 We rewrote this sentence.

Line 165 We deleted “were”.

Line 170 We changed “building” to “buildings”.

Line 174 We changed “presented” to “present”.

Reviewer 3 Report

Authors represented the study of the potential antigens identification in developmental stages of rice weevil, since it may cause a significant allergenic reactions for humans. Overall, this is clear, concise, and well-written manuscript. The introduction is relevant and theory based. The methods are appropriate. The results are clear with adequate explanations. The conclusion should be supplemented with the authors recommendation about the safety measures for workers dealing with stored product. Also, some specific comments are listed within the attached file. After the authors consider the suggestions and make corrections, the paper will be suitable for the publishing in the IJERPH Journal.

Comments and recommendation

Line 23 The methods within the abstract should be change; specifically, the part where the author say that they identify antigens from four life stages of the rice weevil. There are only three stages tested (larvae, pupae and adults). You can say … from three life stages of the rice weevil (larvae, pupae and adults separated by sex) …

Line 47 Add authority after the species name; (L.)

Figure 3 The frequency of the protein fractions 87-89 is incorrect, instead of 97, it should be written 9.7. Please check this value.

Line 146 This sentence should be rephrased; delete “Authors should” or rephrase it.

Line 154/155 When species first mentioned, it should be written in full name along with the authority; it should be written: Sitophilus granarius (L.)

Also, with species Oryzaephilus surinamensis, add authority: (L.)

Line 158 Change S. granaries with S. granaries

Conclusion:

What do the authors recommend for the protection of employees who handle stored products or work in warehouses. Should they wear protective equipment, masks, gloves….?

Give a few sentences about their safety measures in order to avoid allergic reactions to stored product insects.

Reference:

Line 204 Put the full stop after the initials T.M.A.

Line 211 Change “Institute os systematics” with “Institute of systematic”

Line 237/238 The number 1. is redundant

Author Response

Reviewer 3

Thank you very much to the reviewer for constructive comments on our manuscript.

All the remarks from the Reviewer were taken into account.

Line 23 We changed the methods within the abstract. There are only three stages tested (larvae, pupae and adults). We wrote larvae, pupae and adults separated by sex.

Line 47 We added authority after the species name; (L.)

Figure 3 The frequency of the protein fractions 87-89 was incorrect. We changed it.

Line 146 We deleted “Authors should”.

Line 154/155 We wrote full name Oryzaephilus surinamensis and Sitophilus granarius along with the authority.

Line 158 We changed S. granaries with S. granarius.

Line 204 We put the full stop after the initials T.M.A.

Line 211 We changed “Institute os systematics” with “Institute of systematic”

Line 237/238 W deleted the number 1.

According to your suggestion we added in the conclusion a few sentences about their safety measures in order to avoid allergic reactions to stored product insects.

Round 2

Reviewer 1 Report

Dear Authors,

The manuscript has been improved. Yet you need to see again the format of the references as there are missing full stops " . " throughtout them.

Also how did you find the third instar of Sitophilus oryzae? The larvae of this species are inside the wheat kernels,  so you cannot observe the discarded exuviae and therefore cannot count the larval instars. Please provide the appropriate references that prove that the larvae you used were of the trird instar. If not, please do again the experiments providing the correct larval instar.

Best regards

Author Response

Dear Reviewer,

thank you for your valuable attention. One of the co-authors, Prof. Małgorzata Kłyś, for many years has been breeding and studying species of storage insects, including their morphology and biology. Using two dissecting needles, S. oryzae larvae were isolated directly from wheat grains under a magnifying glass in a petri dish. Then, their body length was measured and on this basis, the third larval stage was determined and selected (these two sentences were added to the section materials and methods). I will add that the isolation of the larvae of this insect is complicated and tedious and requires great skills and experience. In addition, to obtain the necessary development stage, much more material must be isolated, which significantly extends the time needed to obtain the necessary amount of research material.

Best Regards